# Dehydration of Xylose to Furfural over Imidazolium-Based Ionic Liquid with Phase Separation

Derun Hua *, Hao Ding, Yunfeng Liu, Jian Li and Baojun Han

Chemical Institute of Chemical Industry, Gannan Normal University, Ganzhou 341000, China; dhao2021@126.com (H.D.); 0900108@gnnu.edu.cn (Y.L.); lijian@sxicc.ac.cn (J.L.); baojunhan@163.com (B.H.)
* Correspondence: huaderun@gnnu.edu.cn

**Abstract:** An environmentally friendly catalyst and task-specific ionic liquid (IL), 1-(4-sulfonic acid) butyl-3-cetyl-2-methyl imidazolium hydrogen sulfate, was applied to the dehydration of xylose to furfural. Its structure was determined by FT-IR, [1]H NMR technologies. The solubility of IL in water changed with the temperature, and had the advantages of homogeneous and heterogeneous catalysts. At the given conditions, xylose conversion of 95.3% and furfural yield of 67.5% were achieved over IL.

**Keywords:** ionic liquid; phase separation; furfural; biomass; xylose

## 1. Introduction

The energy crisis and a concern about the environment have promoted the development of carbon-neutral renewable biomass for energy [1]. Biomass is a sustainable alternative for conventional fossil fuels, particularly when environmentally friendly technologies are applied. Biomass is the only sustainable carbon carrier among renewable sources and can be converted into energy by thermochemical conversion, biochemical conversion, and extraction. Among various conversion methods, hydrothermal conversion is the most promising, without special requirements for materials. Furfural and its derivatives derived from biomass have the potential to make jet and diesel fuel [2], and can be used to synthesize a variety of acids, aldehydes, and alcohols currently obtained from petroleum. Until now, the commercial furfural process has suffered from disadvantages such as furfural loss, equipment corrosion, and high-energy consumption [3]. Solid acid catalysts successfully solved some drawbacks of liquid catalysts, such as separation and recycling et al. [4–6]. Wang et al. applied sulfonated palygorskite solid acid catalyst (PAL-SO₃H) prepared by a mechanochemical process to biomass-derived xylose into furfural, and 87% of furfural yield with 96% selectivity was achieved [7]. Mérida-Morales et al. evaluated zirconium-doped mesoporous silica catalysts in glucose dehydration to 5-hydroxymethylfurfural [8]. In order to minimize furfural degradation reactions, a biphasic system removing furfural from the reaction medium was applied to furfural production from xylose [9]. However, solid catalysts hold poor hydrothermal stability and mass transfer limits, and the cost increases for the application of extractant. How to synthesize a catalyst with advantages of both solid and liquid catalysts is a challenge. Until now, ionic liquids (ILs), which are salts composed of large organic cations and inorganic or organic anions [10], have been used widely in the process of treatment for biomass [11,12]. Some properties of ILs can be tailored by the manipulation of constituent ions, and ILs show excellent catalytic properties. Recently, ILs were applied widely in the conversion of carbohydrates into furfural as solvents [1,13–16]. In some cases, ionic liquids were used as catalysts for the conversion of biomass [17–19]. One of challenges is the separation of ILs in subsequent processes. A key to realize this aim is the structure of IL. Therefore, a form of imidazole-based IL with phase separation at low temperatures and water solubility at high temperatures has been



designed and synthesized, which holds simultaneously the advantages of solid catalyst and liquid catalyst.

The aim of this work is to investigate the dehydration of xylose to furfural catalyzed over IL at low temperatures (393–453 K) and present an environmentally friendly technology for the catalytic dehydration of xylose to furfural. IL is highly acidic, as it contains a sulfonic acid and hydrogen sulfate. IL, in the process, holds performances such as free separation and extractant. The suitable conditions for maximum yield of furfural are presented. A kinetic model is proposed.

## 2. Results and Discussion

ILs were synthesized or purchased in order to determine an effect of the structure of ILs on its water solubility. The structures of ILs were shown in Scheme 1.

**Scheme 1.** The structures of ILs.

ILs with different R (methyl, octyl, dodecyl, cetyl) were synthesized as following Scheme 2, and water solubility tests of ILs were carried out above 70 °C and below 40 °C, respectively. Entry 1 was liquid and water-soluble at room temperature. Entry 2 was semi-solid and fell short of our requirement at room temperature. Entries 3 and 4 were solid, and their properties had marked change. However, Entry 3 dissolved in water below 40 °C, which was outside our requirements. Entry 4 (1-(4-sulfonic acid) butyl-3-cetyl-2-methylimidazolium hydrogen sulfate) met the design requirements and was used in follow-up experiments.

**Scheme 2.** The synthesis routes of imidazolium-based ionic liquid.

Figure 1 showed the FT-IR spectrum of 1-(4-sulfonic acid) butyl-3-cetyl-2-methyl imidazolium hydrogen sulfate. The peaks at 3134 and 3412 cm$^{-1}$ were attributed to the stretching of the hydroxyl group (sulfonic acid group and hydrogen sulfate). The peak at about 2950 cm$^{-1}$ was an asymmetric stretching vibration of the $CH_3$ groups. A strong peak at 2922 cm$^{-1}$ can be related to the (-CH-)$_n$ (n < 4) antisymmetric stretching vibration. The band at 2852 cm$^{-1}$ was related to $CH_3$. C=N vibrations at 1622 cm$^{-1}$ were observed. The peak at 1565 cm$^{-1}$ can be assigned to the imidazolium ring. A strong peak at 1465 cm$^{-1}$ which was assigned the bending vibration of $CH_2$ was found. The characteristic peaks of IL at around 1182 cm$^{-1}$ and 1060 cm$^{-1}$ could be clearly observed in Figure 1, which were ascribed to S=O asymmetric and symmetric stretching vibrations of the -SO$_3$H group, respectively. In addition, S-O vibrations could be observed at 617 cm$^{-1}$ [20]. A peak at around 728 was attributed to (-CH-)$_n$ (n > 4). These special IR peaks indicated that hydrogen sulfate was successfully assembled by 1-(4-sulfonic acid) butyl-3-cetyl-2-methyl

imidazolium molecule as a cation for the corresponding ionic liquid (Entry 4). A detailed analysis of the ¹H NMR spectrum of IL (Figure 2) is listed in Table 2.

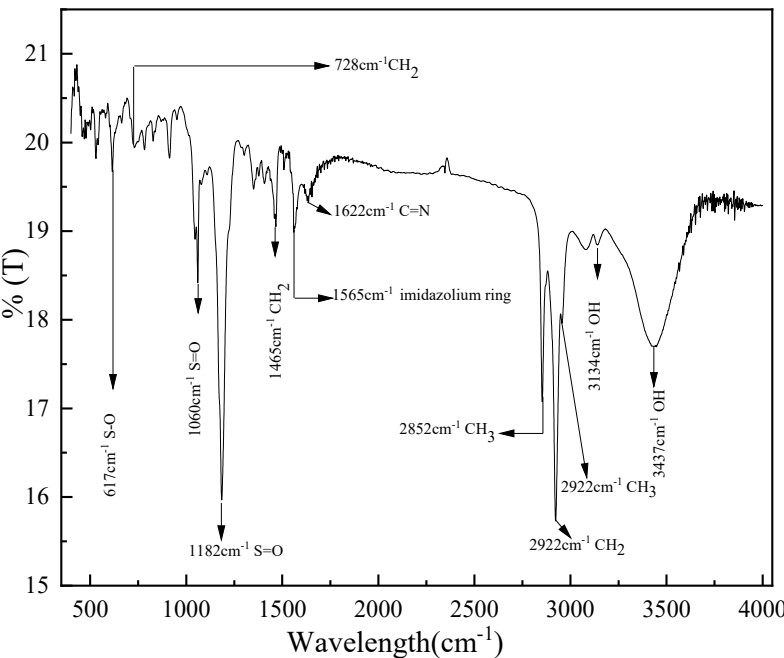

**Figure 1.** FT-IR spectra of 1-(4-sulfonic acid) butyl-3-cetyl-2-methylimidazolium hydrogen sulfate.

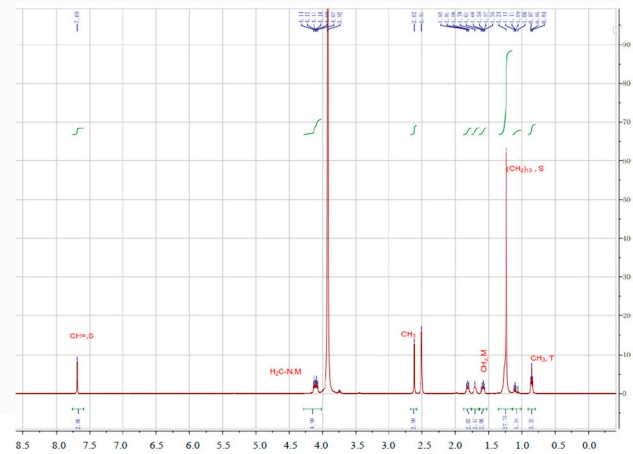

**Figure 2.** ¹H NMR spectra of 1-(4-sulfonic acid) butyl-3-cetyl-2-methylimidazolium hydrogen sulfate (300 MHz, DMSO).

The acid-catalyzed dehydration of xylose to furfural usually proceeds via the consecutive protonation of hydroxyl groups (forming $H_2O^+$ groups) and cleavage of the C-O bonds, leading to the liberation of water molecules (three overall) and formation of trivalent carbocation intermediates which undergo deprotonation, resulting in C=C bond formation [4].

Time, temperature, and catalysts are the key factors for xylose dehydration reaction. In the study, all factors were conducted. The low furfural yield of the dehydration of xylose to furfural in water was reported for the prolonged reaction time due to the formation of byproducts, polymerization, resinification, and condensation reactions, etc. [21]. Thus, the influence of reaction time on xylose conversion and furfural yield was studied between 30–240 min at 160 °C. The results are shown in Figure 3a. It was found that the percentage of xylose conversion and furfural yield increased with the prolonged reaction time, and the maximum yield of furfural (67.5%) was obtained at 180 min with 95.3% conversion. Further, the furfural yield decreased with the increase in reaction time, even though xylose

conversion was 95.3%. The reaction time was above 180 min, and furfural yield in fact decreased with high xylose conversion. A prolonged time promoted the formation of by products and the decomposition of furfural. Thus, higher xylose conversion and lower furfural yield at a longer reaction time were found due to the conversion of xylose into cyclic intermediates or the decomposition of xylose to other byproducts, such as the condensation of furfural [22,23], suggesting that side reactions were dominant. Therefore, the reaction time should not be longer than 180 min. Figure 3b showed the effect of temperature on xylose conversion and furfural yield. It was noted that the maximum conversion and yield were obtained at 160 °C, but the reaction temperature was above 160 °C and furfural yield decreased. The decrease in yield at 160 °C indicated that side reaction rates increased more than that of furfural formation [24]. Subsequently, temperature (160 °C) and time (120 min) were used as reaction conditions. By comparison, the performance of IL was higher than that of high-temperature water and comparable to that of others (Figure 3c). Xylose conversion and furfural yield were only ca. 25% and 15% by HTW (high-temperature water) as a catalyst, respectively. Xylose conversion of ca. 95.3% and furfural yield of ca. 67.5% were achieved over IL, which was comparable to that of liquid acid. The results showed that IL dissolved in water and formed acid solution at high temperature, which catalyzed the dehydration of xylose to furfural. IL showed potential for the dehydration of xylose to furfural. One reason was that IL can provide acidic conditions for the dehydration of xylose. Another possible reason was that IL could play a role as solvent for large bulk, and could also transfer the furfural product from the reaction mixture and prevent furfural dehydration. Recycling of the catalyst was important to develop the industrial catalyst. However, the IL catalyst was used for up to three consecutive recycle runs without regeneration. The results are shown in Figure 3d after three consecutive recycle runs. Results show that there is inconsistency in furfural yield and xylose conversion. The inconsistency in yield and conversion indicate that the IL was unstable. Furfural yield decreased from 65% to 42%, which was attributed to the loss of hydrion. The spent IL was regenerated by solution (4 wt.% $H_2SO_4$), and furfural yield of ca. 62% was achieved (in Figure 2d). The regenerated catalyst complied with the fresh catalyst, thus confirming that the IL catalyst has potential for industry.

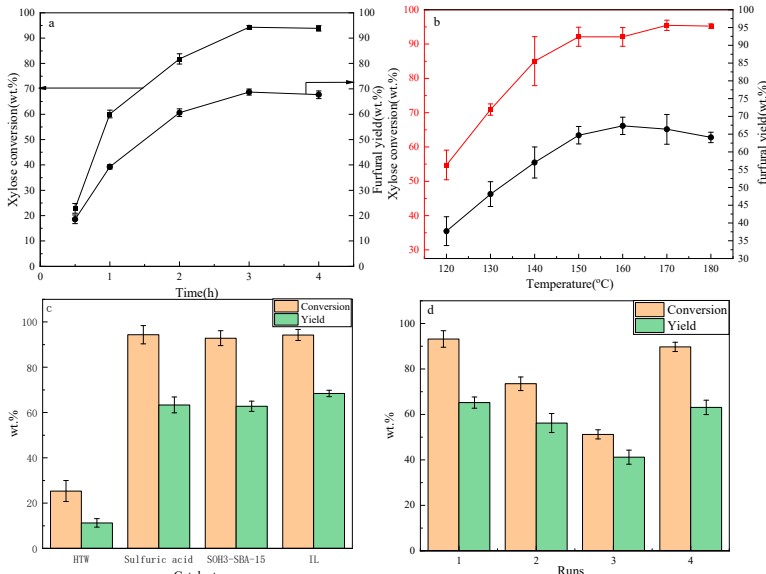

**Figure 3.** Catalytic performance of IL for dehydration of xylose to furfural. Reaction conditions (**a**): *T* = 160 °C, (**b**): *t* = 3 h, (**c**): *t* = 3 h *T* = 160 °C, (**d**): *T* = 160 °C *t* = 3 h, (*C* = 10 wt.% (0.67 mol/L), n (xylose):n (IL) = 5:1).

The kinetic studies in aqueous solution were carried out with IL as a catalyst. The reaction mechanism consisted of furfural formation, furfural loss reactions, and furfural

degradation (Scheme 3). The three reactions were also assumed to be irreversible, and the first reaction (r1) shows the dehydration of D-xylose to furfural, which was assumed to be an irreversible first-order reaction [25,26].

$$R_X = -\frac{d[C]}{dt} = k[C]^n \tag{1}$$

$$k = k_0 e^{-\frac{E_a}{RT}} \tag{2}$$

$$\ln k = -\frac{E_a}{R}\frac{1}{T} + \ln k_o \tag{3}$$

where $R$ is the reaction rate (mol/(L·s), $[X]$ is the xylose concentration (mol/L) and $k$ is the reaction rate constant (s$^{-1}$).

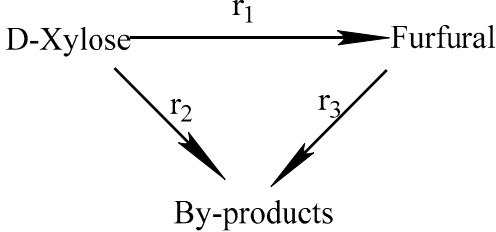

**Scheme 3.** Kinetic mechanism for the conversion of xylose.

In order to evaluate the reproducibility of the data, experiments were repeated three times under the same conditions in Figure 4. The corresponding standard deviations ranged from 2.0% to 7.0%. Therefore, it was concluded that the accuracy of the data might be satisfactory. The research focused on the reactions (r1). The model with first-order kinetics (r1) fitted well with the experimental data. The Arrhenius expression gave a good fit to the rate constants obtained, as illustrated in Figure 3. The corresponding activation energy of 92.1 kJ/mol correlated well with the value obtained by [27]. In Table 1, the Arrhenius parameters are listed for the reaction.

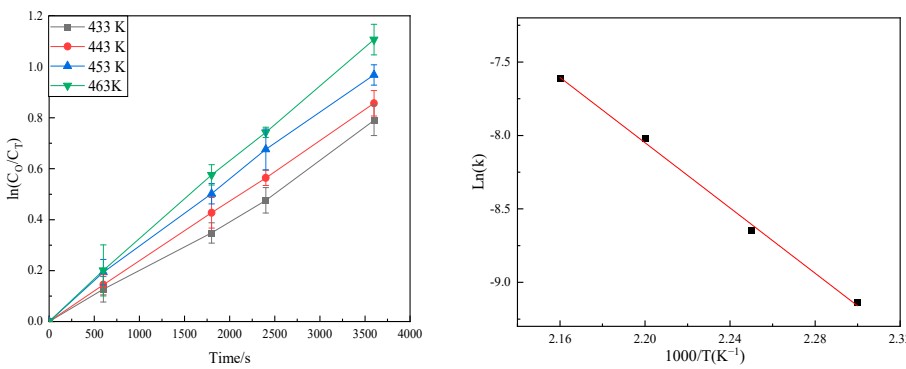

**Figure 4.** Arrhenius plot of k for the temperature range 433–463 K.

**Table 1.** Kinetic parameter of xylose to furfural on IL.

| T/K | K/(mol·L)·s$^{-1}$ | E/kJ·mol | K$_0$/(mol·L) s$^{-1}$ |
|---|---|---|---|
| 433 | $1.09 \times 10^{-4}$ | | |
| 443 | $1.75 \times 10^{-4}$ | | |
| 453 | $2.75 \times 10^{-4}$ | 92.4 | $1.52 \times 10^6$ |
| 463 | $5.00 \times 10^{-4}$ | | |

## 3. Materials and Methods

### 3.1. Materials

1-(4-sulfonic acid) butyl-3-R(methyl, octyl, dodecyl, cetyl)-2-methylimidazolium hydrogen sulfate (95 wt.%) was from Shanghai Mooney Chemical Technology Co., Ltd. (Shanghai, China). All chemicals used were from Sigma-Aldrich (Shanghai, China), as follows: Xylose ($\geq$99.0 wt.% purity), furfural ($\geq$99.0 wt.% purity).

### 3.2. Instruments

The target IL was characterized by FT-IR, $^1$H NMR. The spectra ranging from 400 to 1300 cm$^{-1}$ were collected on the Digilab FTS-3000 FT-IR spectrometer using KBr pellet technique. IL was determined by Avance III 300 MHz $^1$H-NMR spectroscopy (Bruker Company, Karlsruhe, Germany).

### 3.3. Synthesis of Catalysts

ILs were synthesized or purchased. Reaction of neutral nucleophile imidazole derivates (0.3 mol) with 1,4-butane sultone (0.3 mol) formed the requisite zwitterions. Subsequently, the synthesized solid was washed repeatedly with ether and was filtrated to remove nonionic residues and dried under a vacuum. Then, sulfuric acid (98%, 16.4 mL) was added dropwise to a 200 mL three-port conical flask containing the solid at 0 °C. The mixture was stirred at 80 °C for 10 h. Product was washed with diethyl ether and dried in vacuo at 40 °C for 4 h to obtain the armed product. We synthesized various ionic liquids in order to verify the effect of branched chains on their water solubility, and screened ionic liquids that met the requirements (See Table 2).

**Table 2.** $^1$H NMR data and analysis.

| No. | Chemical Shift/δ | Identification for the Hydrogen Spectra | Peak Profile | Integral Area |
|---|---|---|---|---|
| 1 | 0.84–0.87 | -CH$_3$ | T | 3.33 |
| 2 | 1.24 | (-CH$_2$-)$_{13}$ | M | 27.76 |
| 3 | 1.55–1.61 | -CH$_2$- | M | 2.06 |
| 4 | 2.62 | -CH$_3$ | S | 2.99 |
| 5 | 4.07–4.14 | H$_2$C-N | M | 5.00 |
| 6 | 7.69 | HC= | S | 2.00 |

### 3.4. Reactions

In the reaction, xylose (2.5 g), catalyst (0.5 g), and distilled water (25 mL) were poured into a reactor (autoclave); xylose concentration and the ratio of xylose to IL was 10 wt.% and 5, respectively. The reactor was heated to a temperature and kept for the set time. The composition of aqueous phase was determined on an LC-20AD (Shimadzu, Japan) equipped with a RID-10A refractive index detector and a Bio-rad Aminex HPX-87H column (300 × 7.8 mm). The conversion of xylose, selectivity of furfural, and yield of furfural were estimated by the following equations:

$$Con_{xylose} = \frac{(m_{xylose-s} - m_{xylose-e})\big/ M_{xylose}}{m_{xylose-s}\big/ M_{xylose}} \times 100\% \qquad (4)$$

$$Sel_{furfural} = \frac{\left(m_{furfural-water}\right)/M_{furfural}}{\left(m_{furfural-s} - m_{furfural-e}\right)/M_{furfural}} \times 100\% \qquad (5)$$

$$Y_{furfural} = Con_{xylose} \times Sel_{furfural} \qquad (6)$$

## 4. Conclusions

The dehydration of xylose to furfural in aqueous solution by acidic ionic liquid (1-(4-sulfonic acid) butyl-3-cetyl-2-methyl imidazolium hydrogen sulfate) catalyst was studied in a temperature range of 433–463 K. The results indicated that the catalyst holds the characteristics of easy separation and high-mass transfer, and exhibits a great potential for the replacement of conventional catalysts. The excellent catalytic performance of IL was attributed to the strong Bronsted acidity of IL (double position acids, sulfonic acid, and hydrogen sulfate). A first-order kinetic model with an activation energy of 92.4 kJ/mol was observed. A suitable temperature was found to be 433 K, where a balance between the rate of formation of furfural and the degradation of furfural was achieved. At this temperature, a $SO_3H$-functionalized ionic liquid exhibited high catalytic activity for the synthesis of furfural from xylose (xylose conversion of 95.3% and furfural yield of 67.5%).

**Author Contributions:** Conceptualization, D.H., J.L. and H.D.; methodology, B.H.; software, Y.L.; validation, D.H. and B.H.; formal analysis, D.H. and J.L.; investigation, D.H.; resources, D.H.; data curation, H.D.; writing—original draft preparation, D.H. and H.D.; writing—review and editing, D.H.; visualization, B.H.; supervision, D.H.; project administration, D.H.; funding acquisition, D.H. All authors have read and agreed to the published version of the manuscript.

**Funding:** This research was funded by the National Natural Science Foundation of China, grant number 21466001 and Open Foundation of Key Laboratory of Jiangxi University for Functional Materials Chemistry (FMC17301).

**Institutional Review Board Statement:** Not applicable.

**Informed Consent Statement:** Not applicable.

**Data Availability Statement:** The data used to support the findings of this study are available within the article.

**Conflicts of Interest:** All authors declare no conflict of interest.

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
