# Peer review of "Dehydration of Xylose to Furfural over Imidazolium-Based Ionic Liquid with Phase Separation"

_catalysts, doi:10.3390/catal11121552_

Round 1

Reviewer 1 Report

The authors complied with a adequate percentage of the comments. The manuscript is still too short. I recommend publishing as a communication.

Reviewer 2 Report

The manuscript reports the use of functionalized imidazolium ionic liquids (ILs) as catalysts for the degradation of D-xylose. The particular benefit is that the side-chain length of the IL cation can be utilized to tailor the properties of the catalyst. Overall, I think this manuscript would certainly be interesting for the community, however, the presented data/figures/equations appear somewhat careless. Therefore, I would consider to publish the manuscript after addressing the following minor issues:

  • Although it is rather obvious, the variables in Eqs. 1-3 should be specified in the text for a complete definition.
  • Scheme 2: The chemical structure of 1,4-butane sultone is incorrect.
  • Figure 2: The peak labels & integrals as well as axis labels are rather small. The authors should consider to enlarge them.
  • Below scheme 3: The equation labels 1-3 are reused, this should be avoided. Moreover, the exponent 'n' could be omitted when assuming a first-order reaction.

Author Response

This manuscript is a resubmission of an earlier submission. The following is a list of the peer review reports and author responses from that submission.

Round 1

Reviewer 1 Report

This study presents an environmentally friendly technology for the catalytic dehydration of xylose to furfural. A sensitivity analysis was performed on the kinetic model. Some very important comments are given below:

1) Authors should further emphasize on the novelty of their work.

2) Some minor typos, grammar and syntax errors should be carefully revised and corrected accordingly.

3) Reference can be even more updated (more recent relative works).

4) Scheme 1 is placed just afrer "Results and Discussion". This is wrong and should be placed after the appropriateb discussion in the text.

5) Fig1: The label of x-axis is missing. Only units were given.

6) Fig2: It is completely misreadable.

7) The major drawback of this work is the very poor discussion. 19 Refs are cited in Introduction and none in Results and Discussion part. Authors should re-write this section. 

8) The work evnthough it is "article" is very short. It should be drastically enriched.

Reviewer 2 Report

The manuscript presents a rather clever conception for the conversion of xylose to furfural via the catalytic mediation of an imidazolium based ionic liquid. It does not contain vital mistakes but vital information is missing. I recommend resubmission of a far more detailed version.

Introduction is too short. A paragraph introducing the problem; a second presenting the background of catalysts used for the transformation of xylose to furfural a third about ionic liquids and a fourth describing the current project with more details would be sufficient.

Page 1 Lines 21—22 “Now, commercial furfural process suffers from disadvantages such as furfural loss, equipment corrosion, and high-energy consumption.” Please add references to these disadvantages of the current state of the art.

Experiment section should be changed to Materials and Methods according to the Instructions for Authors.

The materials, the instruments, the synthesis of the catalysts and the catalytic transformation reaction should be four separate entities. (Paragraphs or even subchapters)

The octyl and dodecyl imidazole derivatives are missing from the materials section.

IR and NMR spectrophotometers are missing from the instruments section

In the IR spectrum the peaks at 2852 and 2922 cm-1 correspond to the symmetric and asymmetric stretching vibrations of the CH2 groups. The respective asymmetric stretching vibration of the CH3 groups is visible. It is the one beside the peak at 2922 cm-1 at about 2950 cm-1. The symmetric peak is most probably the shoulder beside the peak at 2852 cm-2.

In the NMR spectrum the ppm of the x axis, are unreadable.

The detailed analysis of the NMR spectrum of IL is not so detailed. The peaks corresponding to the γ-CH2 to the imidazolium nitrogen of the cetyl chain; to the β-CH2 to the imidazolium nitrogen of the long aliphatic chain;  to the β-CH2 to the imidazolium nitrogen of the butyl aliphatic chain and to the β-CH2 to the sulfonic acid group are not mentioned. In addition, the strong peak at about 3.9 is due to moisture? Any idea for the upshift from 3.33. The peak at 1.25 ppm cannot be singlet it is multiplet.

The authors claim in the experimental section that the composition of aqueous phase was determined on a LC-20AD (Shimadzu, Japan). HPLC, I suppose. Where is the diagram that proves the selective conversion to furfural? Where is the NMR spectrum of furfural? What are the byproducts? Again, NMR characterization would be useful.

Round 2

Reviewer 1 Report

1) As I wrote in the 1st round

"The major drawback of this work is the very poor discussion. 19 Refs are cited in Introduction and none in Results and Discussion part. Authors should re-write this section". Authors just put 4 Refs with almot zero discussion. I am sorry to say that, but this cannot be accepted.

2) The paper is "Article" as authors stated and not communication. Eventhough it is communication, the discussion is very poor.

I recommend to re-write the paper.

Reviewer 2 Report

The authors made substancial efforts to comply with my comments. The manuscript is indeed suited to be published as communication if this format is acceptable by the editors.